# Atmospheric pollution in Ulaanbaatar: Persistence and long-run trends

**Ariundelger Ariunsaikhan[1], Chultem Batbold[2], Sonomdagva Chonokhuu[2], Luis Alberiko Gil-Alana[3]\***

1 Social Science of Doctoral School, University of Lodz, Poland, 2 School of Engineering and Technology, National University of Mongolia, Ulaanbaatar, Mongolia, 3 University of Navarra, NCID, DATAI, Pamplona, Spain and Universidad Francisco de Vitoria, Madrid, Spain

\* alana@unav.es

## Abstract

This paper investigates the presence of long-run trends and persistence in various pollutants in the city of Ulaanbaatar, Mongolia using fractional integration. Using daily data from January 1st, 2022 until May 31st, 2024 we investigate the statistical properties of four pollutants, namely, $NO_2$, $SO_2$, $PM_{10,}$ and $PM_{2.5}$. The results indicate the presence of significant negative time trends in the cases of $SO_2$ and $PM_{2.5}$ and evidence of long memory and mean reverting patterns in all four pollutants. Policy implications of the results obtained are reported at the end of the paper.

## 1. Introduction

Mongolia, a landlocked Central Asian country, has unique ecosystems and cultures. Its vast steppes and deserts experience about 250 sunny days yearly [1]. In recent years, rapid urbanization has taken place. According to the National Statistic Committee, as of 2022, approximately half of Mongolia's population resides in Ulaanbaatar (UB), the capital, with 53% of these residents living in "ger areas" (where households live in houses and gers, Mongolian traditional yurts, without any central heating system) [2]. Residents in "ger areas" of the capital typically use coal briquettes for heating, whereas those in rural areas continue to rely on raw coal due to the harsh wintertime. Several factors contribute to air pollution in UB. Geographically, UB is the world's coldest capital [3], necessitating significant heating during the winter. Additionally, the city is situated in a valley surrounded by mountains [4], which, combined with specific meteorological conditions [5], exacerbates air pollution. From the socio-economic perspective, rapid urbanization and the reliance on specific energy sources also play a major role [6].

In 1990, Mongolia, the world's second-oldest communist state, began fundamentally transforming its economy and swiftly transitioned to a multi-party democracy [7]. Since then, Mongolia has evolved into a vibrant democracy, with its GDP per capita tripling [8]. In 2023, mineral products comprised 84.1% of Mongolia's total exports,

**Data availability statement:** All relevant data are within the paper and its Supporting Information files.

**Funding:** Luis A. Gil-Alana gratefully acknowledge financial support from the Grant PID2023-149516NB-I00/ AEI/10.13039/501100011033/ FEDER, UE

**Competing interests:** The authors have declared that no competing interests exist.

while non-mineral products accounted for 15.9%. The main export destinations were China, Switzerland, and South Korea. On the import side, mineral products constituted 24.9% of imports, followed by road and air vehicles and their parts at 21.5%, and mechanical equipment, electrical appliances, and spare parts at 16.1%. These imports were primarily from China, Russia, and Japan [9]. However, the sources of air pollution in UB are not from mining; instead, they are primarily from households, power plants, and transportation [10–13].

According to [14,15], Mongolia's annual average $PM_{2.5}$ concentration, weighted by population, was 46.6 µg/m$^3$, ranking 4th in the world in 2020 [14]. By 2023, this figure had decreased to 22.5 µg/m$^3$, placing Mongolia 39th globally [15], which is still 4.5 times higher than the WHO air quality guideline [16]. Note, however, that the IQAir website contains fewer than 10 monitoring points throughout Mongolia, which may be not sufficient for the whole territory. It is important to note that this report focuses solely on $PM_{2.5}$ concentration. However, other pollutants, such as $SO_2$, have been increasing; for instance, the annual $SO_2$ concentration rose from 50 µg/m$^3$ in 2020–70 µg/m$^3$ in 2022. Conversely, the $NO_2$ concentration has remained stable at 40 µg/m$^3$ [2], four times higher than the WHO air quality guideline. A study by [17] indicates a high prevalence of persistent cough symptoms among schoolchildren in urban and suburban districts of UB. The research found that outdoor $SO_2$ concentrations were linked to persistent cough symptoms, while $NO_2$ concentrations were associated with current wheezing symptoms in children [17].

This study examines the statistical properties of four primary air pollutants in Ulaanbaatar, Mongolia. This work has two main objectives: first, to determine if time trends are present in the data to show potential decreasing trends in its temporal evolution. Second, we use a specific model named fractional integration, with which, due to its characteristics, we can determine the degree of persistence in the data with a single parameter (the order of integration) and whether shocks in the series will have transitory or permanent effects. Our results indicate that the four pollutants examined are mean reverting with shocks having transitory effects, and significant negative trends are found in the cases of $SO_2$ and $PM_{2.5}$.

The rest of the paper is structured as follows: Section 2 presents a literature review on atmospheric pollution issues, with a focus on time series analysis; Section 3 displays the data and the methodology used in the paper; Section 4 shows the empirical results, while Section 5 contains the discussion and Section 6 concludes the paper.

## 2. Literature review on atmospheric pollution

This study focuses on primary pollutants in Mongolia, where coal combustion is the primary source of air pollution due to its widespread use in heating systems. Other studies similar to ours either focus on analysing the spatiotemporal characteristics of primary air pollutants or on assessing air pollutants focused on these four pollutants. For instance [18], evaluated the effectiveness of energy policies using spatial econometric methods, focusing on $PM_{10}$, $PM_{2.5}$, and $SO_2$ [19]. examined major air pollutants' spatiotemporal and sectoral distribution and their drivers, selecting $NO_x$, $SO_2$, and

dust as key pollutants. Similarly [20], focused on $SO_2$, $NO_2$, and PM in India, while [21] analysed $PM_{2.5}$ and $NO_2$ using city-wide air monitoring data from New York, USA. Other studies have adopted geospatial analysis to assess pollution trends [22]. monitored $SO_2$, $NO_2$, and PM using GIS modelling in India [23]. Analysed monthly data from five different monitoring stations in India, covering $NO_2$, $SO_2$, and $PM_{10}$ from 2011 to 2020 [24]. investigated the spatiotemporal characteristics of $PM_{2.5}$, $PM_{10}$, $SO_2$, and $NO_2$ across 11 stations in China. Other relevant studies include [25], who analysed the spatial distribution of $PM_{2.5}$ and $PM_{10}$ in Addis Ababa, Ethiopia [26]. focused on identifying the main contributors to air pollution in Beijing, specifically analysing $SO_2$ levels [27]. Sharma et al. (2019) examined air pollution trends across various geographical locations in India, considering $SO_2$, $NO_x$, and $PM_{2.5}$. Additionally [28], updated previous marine air pollution estimations by analysing emissions of $NO_x$, $SO_x$, and $PM_{2.5}$.

Focussing on the air pollution in Mongolia [29], concluded that air pollution, specifically $SO_2$ concentration, has worsened due to rapid urbanisation and industrialisation since the mid-1990s. $SO_2$ concentrations were obtained from the Central Laboratory of Environmental Monitoring (CLEM) under NAMHEM, which operates Mongolia's air quality monitoring network. This study used $SO_2$ data and meteorological parameters with at least 70% data completeness, recorded from 14 sites—including the capital and provincial centres—between January 1, 1996, and December 31, 2009. Pollution is found to be particularly severe in urban areas, near steel industry sites, and during winter. It is noticeable that in UB, $SO_2$ concentrations increase with decreasing wind speed and temperature, as well as with increasing relative humidity. Meteorological parameters and emissions from industrial sources and gers to the north of the city seem to influence the dispersion of $SO_2$ in UB [30]. focused on $PM_{10}$ and $PM_{2.5}$, which are the primary pollutants. However, they estimate the air quality index (AQI) for major pollutants such as $PM_{2.5}$, $PM_{10}$, $SO_2$, $NO_2$, CO, and $O_3$. The daily monitoring data was obtained from the Mongolian Ministry of Nature, Environment, and Tourism; the hourly monitoring data for $PM_{2.5}$ and $PM_{10}$ were obtained from the OpenAQ website. The meteorological data was obtained from the National Oceanic and Atmospheric Administration.

According to researchers, the $PM_{2.5}/PM_{10}$ ratio has decreased yearly, with the highest concentrations in winter and the lowest in summer in UB. Daily PM concentration showed a bimodal pattern: it decreased during the day and slightly increased in the afternoon due to temperature inversion. PM concentrations were significantly higher during the heating season, indicating coal-fired heating as the main cause of air pollution in UB.

[31] identified sources in UB city by characterising polycyclic aromatic hydrocarbons in total suspended particles. Researchers collected samples from five locations using a high-volume air sampler for 3–24 hours. Concentrations varied across different sites and seasons, with the highest in the ger area during winter. The main pollution source in the city centre during winter was vehicle emissions, while other sites showed mixed contributions from coal, petroleum, and other biomass combustion [32]. suggested some recommendations for further research to improve air quality monitoring. One key recommendation in the paper is the establishment of more air quality monitoring sites and the enhancement of air quality investigations which this study addresses.

From the methodological side, several papers have employed fractional integration in air quality analysis. Thus, for example [33], used a fractionally integrated framework to analyze U.S. air quality using datasets retrieved from the EPA database. This allowed fractional differentiation degrees for stationary I(0) series, offering more flexibility in the dynamic data specification. Additionally [34], I investigated the time trends and persistence of $PM_{2.5}$ in 20 megacities. They employed techniques based on long-range dependence or long memory, with a focus once again on fractional integration models using daily average data taken from the World AQI. The advantage of using a long-memory fractionally integrated framework in analysing air quality time series data lies in its ability to capture complex and persistent patterns in the data. This methodology allows for a more nuanced understanding of the dynamics of air quality variables, such as particulate matter pollution, by considering fractional values in the degree of differentiation. Other papers using a similar methodology in the analysis of air pollution include [35] in four Chinese cities [36], in ten European capitals and [34] in the case of London. Alternative methodological approaches such as AutoRegressive Integrated Moving Average (ARIMA) have been

employed in the analysis of Indian air quality data by [37–39] e3s 021), while [40] and [41] use a hybrid deep learning/ Kriging model and a new complex-network-based model respectively. A Simple Linear Regression (SLR) model was used by [42] to evaluate the accuracy of five low-cost air quality sensors against a particulate reference analyzer. Their findings suggest that low-cost sensors are unreliable for accurately measuring air quality in indoor environments. Various studies have also applied machine learning (ML) and artificial neural network (ANN) models [43]. introduced a location-invariant air pollution prediction model with strong geographic generalizability, integrating light gradient-boosted regression (GBR) within a ML framework. Experiments on diverse datasets demonstrated superior performance to standard forecasting methods, such as recurrent neural networks and transformers. An interpretability analysis identified key factors influencing air pollution levels and revealed geographical patterns of high pollutant concentrations. Furthermore [44], predicted the concentrations of six major air pollutants using ML and deep learning techniques, while [45] applied an ANN model to predict dust concentration in India's deepest opencast copper mines, achieving strong agreement between observed and predicted data.

## 3. Data description and methodology

### Data collection and air quality monitoring

The National Agency for Meteorology and Environmental Monitoring (NAMEM) at 19 monitoring locations officially measured the air pollution pollutants in UB, Mongolia. The daily data used in this study was obtained from NAMEM. In Mongolia, air quality monitoring follows WHO guidelines; however, not all pollutants are consistently measured. Some monitoring stations record only three pollutants—$PM_{10}$, $SO_2$, and $NO_2$—while others measure four, depending on equipment availability. Only a few monitoring stations measure all six major pollutants. Due to these limitations, we selected a single monitoring site for representing the UB. This site, located in central Ulaanbaatar, measures four pollutants: $PM_{10}$, $PM_{2.5}$, $SO_2$, and $NO_2$.

### Data description

Table 1 briefly describes the dataset covering January 2022 to June 2024 for four different types of air pollutants. Over the past two years, the average concentrations were as follows: $SO_2$ at 31.9±30.6 µg/m³, $NO_2$ at 80.3±28.3 µg/m³, $PM_{10}$ at 87.8±49.6 µg/m³, and $PM_{2.5}$ at 36.8±32.9 µg/m³ (See Table 1).

Fig 1 illustrates the historical data for four different pollutants over the last two years. The general trend shows that $SO_2$, $NO_2$, and $PM_{2.5}$ levels increased during the winter, while $PM_{10}$ levels increased in autumn and spring.

### Methodology

Dealing with the methodology, we use a particular type of long memory model widely used in the context of environmental studies and denominated fractional integration. The idea that is behind this concept is that the number of differences required in a series over time to render it stationary I(0) may be a non-integer positive real value. In other words,

**Table 1. Descriptive statistics.**

| Series | Mean (µg/m³) | Std. Deviation (µg/m³) | Maximum Val. (µg/m³) | Minimum Val. (µg/m³) |
|---|---|---|---|---|
| $SO_2$ | 31.9 | 30.6 | 172 | 0 |
| $NO_2$ | 80.3 | 28.3 | 189 | 10 |
| $PM_{10}$ | 87.8 | 49.6 | 474 | 2 |
| $PM_{2.5}$ | 36.8 | 32.9 | 203 | 1 |

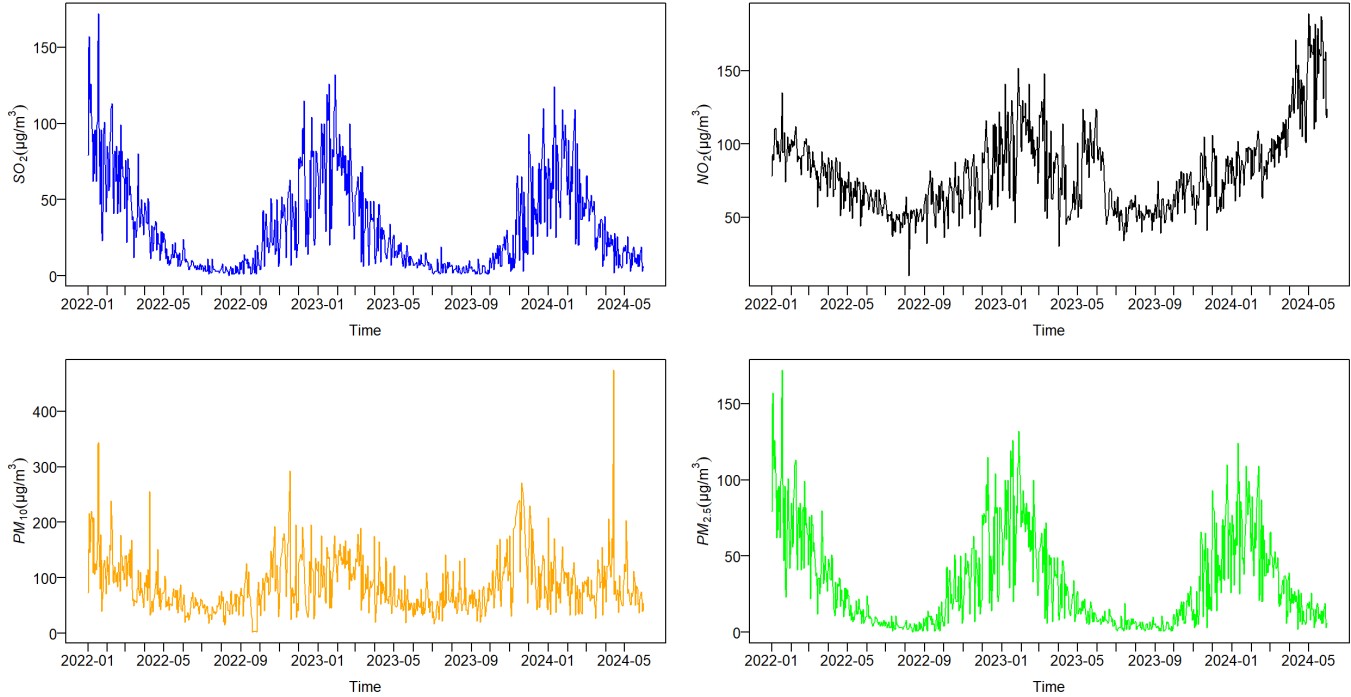

**Fig 1. Historical data of SO$_2$, NO$_2$, PM$_{10}$ and PM$_{2.5}$ from January 2022 to May 2024.**

a process {x(t), t = 0, ± 1, …} is said to be fractionally integrated or integrated of order d, and denoted as I(d) if it can be represented as:

$$(1-B)^d x(t) = u(t), \quad t = 0, \pm 1, \ldots \tag{1}$$

where B is the backshift operator, such that B$^k$x(t) = x(t-k) and u(t) is I(0) or integrated of order 0, which means that it is a second order (or covariance) stationary process with a spectral density function that is bounded and positive at all frequencies. Within this category, the simplest process is the white noise, characterized by displaying a zero mean, a constant variance and no autocorrelation, though it also permits serial correlation like that of the stationary Autoregressive Moving Average (ARMA) process.

The differencing parameter d is crucial to determine if shocks in the series have transitory or permanent effects. Thus, if d is smaller than 1, shocks are expected to be transitory and the recovery is faster the lower the value of d is. On the other hand, values of d equal to or superior to 1 indicate a lack of mean reversion and thus permanency of shocks. Moreover, the fact that d is a real value permits us to consider a wide range of alternatives, including among others,

i)   short memory or I(0) processes, if d = 0,

ii)   stationary long memory I(d) processes, if 0 < d < 0.5,

iii)   nonstationary though mean reverting I(d) processes, if 0.5 ≤ d < 1,

iv)   unit roots or I(1) processes, if d = 1, and

v)   explosive patterns, if d > 1.

Of particular interest in the present work are the process in iii) since being nonstationary, they still display transitory shocks though with long lasting effects. In the empirical application conducted in the following section, we are also interested in the potential presence of trends in the data. Thus, we assume now that x(t) in (1) are the errors in a regression that includes a constant and a linear time trend, i.e.,

$$y(t) = \alpha + \beta\, t + x(t), \quad t = 1, 2, \ldots, \tag{2}$$

where y(t) is the series corresponding to the observed data; α and β are the unknown constant and trend respectively, and x(t) follows Equation (1).

The estimation is conducted via the log-likelihood function, following a testing procedure developed in [46] and widely used when analyzing univariate time series data. Its functional form can be found in [47], and applications in environmental science using this approach are, among others, [34–36,48,49],.

This method is a testing procedure. The null hypothesis is:

$$H_o : d = d_o, \tag{3}$$

which is tested in the model given by Eqs. (1) and (2) for any real value $d_o$. It is based on the Lagrange Multiplier (LM) principle and it has a standard normal null and local limit distributions. This allows us to consider a confidence band for the non-rejection values of d. In addition, this standard limit behaviour holds whether or not deterministic terms (like those in (2)) are included in the model, and it is supposed to be the most efficient method in the Pitman sense against local alternatives. (See, [46]. …"

## 4. Empirical results

The model under investigation is the one given by Equations (1) and (2), i.e., )

$$y(t) = \alpha + \beta\, t + x(t), \quad (1 - B)^d x(t) = u(t), \tag{4}$$

with x(t) = 0 for t ≤ 0. Therefore, there are two main parameters of interest. On the one hand, β, since a significant value of this parameter will support the existence of a time trend in the data. On the other hand there is the differencing parameter, d, indicating the degree of persistence of the series.

Tables 2 and 3 refer to the case where u(t) is uncorrelated, with zero mean and constant variance. Tables 4 and 5 report the results with serial correlation following the exponential spectral model of [50]. This is a non-parametric method that approximates the behavior of AutoRegressive (AR) structures by means of a spectral density function with an exponential form. This model accommodates very well to the functional form of the test statistic of Robinson (1994) used in this application (see,.e.g., [51].

**Table 2. Estimates of d: White noise errors.**

| Series | No terms | An intercept | An intercept with a linear time trend |
|---|---|---|---|
| $SO_2$ | 0.589 (0.548, 0.633) | 0.542 (0.504, 0.587) | **0.551 (0.514, 0.593)** |
| $NO_2$ | 0.620 (0.584, 0.674) | **0.567 (0.532, 0.611)** | 0.564 (0.528, 0.610) |
| $PM_{10}$ | 0.486 (0.443, 0.534) | **0.447 (0.401, 0.497)** | 0.449 (0.402, 0.498) |
| $PM_{2.5}$ | 0.531 (0.492, 0.574) | 0.501 (0.467, 0.543) | **0.505 (0.471, 0.542)** |

The values are the estimates of d and in parenthesis appear the 95% confidence bands. We report in bold the selected deterministic case for each series.

**Table 3. Estimated coefficients: White noise errors.**

| Series | d (95% conf. interval) | Intercept (t-value) | Time trend (t-value) |
|---|---|---|---|
| $SO_2$ | 0.542 (0.504, 0.587) | 86.260 (7.82) | -0.0744 (-2.27) |
| $NO_2$ | 0.564 (0.528, 0.610) | 87.246 (96.91) | ----- |
| $PM_{10}$ | 0.447 (0.401, 0.497) | 106.543 (6.40) | ----- |
| $PM_{2.5}$ | 0.505 (0.471, 0.542) | 69.351 (5.72) | -0.0494 (-1.97) |

The values in column 2 are the estimates of d and the 95% confidence bands. Columns 3 and 4 report the estimates of the intercept and the time trend with their corresponding t-values. --- means lack of statistical significance.

**Table 4. Estimates of d: Autocorrelated errors.**

| Series | No terms | An intercept | An intercept with a linear time trend |
|---|---|---|---|
| $SO_2$ | 0.493 (0.442, 0.536) | 0.461 (0.421, 0.506) | **0.481 (0.444, 0.531)** |
| $NO_2$ | 0.583 (0.544, 0.643) | **0.533 (0.491, 0.581)** | 0.530 (0.490, 0.578) |
| $PM_{10}$ | 0.446 (0.372, 0.504) | **0.393 (0.337, 0.463)** | 0.401 (0.338, 0.472) |
| $PM_{2.5}$ | 0.502 (0.460, 0.573) | 0.482 (0.432, 0.531) | **0.485 (0.441, 0.539)** |

The values are the estimates of d and in parenthesis appear the 95% confidence bands. We report in bold the selected deterministic case for each series.

**Table 5. Estimated coefficients: Autocorrelated errors.**

| Series | d (95% conf. interval) | Intercept (t-value) | Time trend (t-value) |
|---|---|---|---|
| $SO_2$ | 0.481 (0.444, 0.531) | 79.149 (8.61) | -0.0678 (-3.02) |
| $NO_2$ | 0.533 (0.491, 0.581) | 87.514 (10.64) | ----- |
| $PM_{10}$ | 0.393 (0.337, 0.463) | 101.984 (7.93) | ----- |
| $PM_{2.5}$ | 0.485 (0.441, 0.539) | 68.478 (5.97) | -0.0488 (-1.91) |

The values in column 2 are the estimates of d and the 95% confidence bands. Columns 3 and 4 report the estimates of the intercept and the time trend with their corresponding t-values. --- means lack of statistical significance.

Starting with the results based on white noise errors we observe in Table 2 that the time trend coefficient is statistically significant in the cases of $SO_2$ and $PM_{2.5}$. For the other two pollutants, however, the intercept is the only required deterministic term. Moreover, in the cases of $SO_2$ and $PM_{2.5}$ the slope is negative, implying a long-term decrease in the number of these types of emissions, with the decrease being higher in case of the $SO_2$ (see Table 3). Dealing now with the degree of persistence, i.e., the value of the differencing parameter, we notice that in the four series the value is within the interval (0, 1) supporting the hypothesis of a long memory pattern. This value is lower than 0.5 in case of the $PM_{10}$, and slightly higher for the other three pollutants. Nevertheless, the confidence intervals include values below and above this number, supporting thus stationary and nonstationary hypotheses in all cases. More importantly, the values are all strictly below 1, indicating mean reversion and transitory effects of the shocks.

We next conduct the same type of analysis but based on autocorrelated errors. Table 4 reports the values of d once again for the three cases of i) no terms, ii) an intercept, and iii) an intercept with a linear time trend, while Table 5 focuses on the estimated values for the selected specifications.

The results are very similar to the previous case and based on white noise errors. Thus, the time trend is significantly negative in the cases of $SO_2$ and $PM_{2.5}$ while it is statistically insignificant in the other two cases, $NO_2$ and $PM_{10}$. Looking at the values of d, they are smaller than in the previous case though still within the interval (0, 1) and supporting the hypothesis of long memory and transitory shocks. These values are 0.293 for the $PM_{10}$; 0.481 for $SO_2$; 0.485 for $PM_{2.5}$ and 0.533 for $NO_2$. Thus, the four pollutants display again a mean reverting pattern.

## 5. Discussion

According to the results reported in this work, the trend for $PM_{2.5}$ and $SO_2$ concentrations significantly reduced in the long term. Numerous factors affect air pollution, such as the population, overall vehicle numbers, industrial activity, number of power plants, metrological conditions, and GDP output. For the specific case of Ulaanbaatar, the population has increased steadily year by year with a 1.6% annual rate of change (U[52, 53] and the number of vehicles has increased significantly over the past decade [54,55]. This growth has led to numerous changes in the city's traffic patterns, such as delays in daily commuting, severe congestion on main arterial roads [56], and increased emissions. Specifically, the number of trucks has grown by a multiple of 1.57, buses by 3.9, passenger cars by 1.82, and vehicles older than 10 years by a ultiple of 1.83 [57]. Despite the increasing population and vehicles, a long-term decrease in $SO_2$ and $PM_{2.5}$ levels might be attributed to major initiatives by government and non-government organizations. These initiatives include urban planning improvements, reforms in the transport sector, adopting renewable energy sources, and replacing coal usage, which have all shown positive results. However, it is too early to identify the factors contributing to these lower levels definitively. Further research is required to accurately assess various influencing factors, such as social, economic, and climatic conditions.

From Table 4, $NO_2$ has the highest intercept, while $PM_{10}$ has the lowest. Also, $NO_2$ displays the highest degree of persistence, as measured by d, and PM10 displays the lowest. This indicates that in the event of a shock, $NO_2$ levels will take longer to return to normal, whereas $PM_{10}$ levels will stabilize more quickly. We have investigated the reasons for this result and have tried to explain it based on the physical and chemical characteristics of the pollutants, and their sources. With regard to the characteristics of pollutants, fine particles are found to be transported farther from Ulaanbaatar than gases [58] due to the longer atmospheric lifetime in which gas pollutants can undergo chemical transformations, becoming part of secondary pollutants. However [59], concluded that larger particles typically settle within shorter distances from the source due to gravitational forces. Also, as previously mentioned, the increasing number of vehicles in Ulaanbaatar is one of the major contributors to the high concentrations of $NO_x$ and PM. According to [60], the total emissions on the main roads were estimated by the hour, day, and year, and analyzed for each vehicle type, age, and link road. The annual concentration of $NO_x$ was 6905.7 tons, and PM was 301.7 tons. Regarding vehicle type, trucks and buses accounted for 49% and 34% of $NO_x$ concentration and 30% and 15% of PM concentration, respectively. Passenger cars were responsible for 17% of $NO_x$ and 55% of PM concentrations. Concerning vehicle age, vehicles older than 10 years accounted for 96% of $NO_x$ and 82% of PM concentration, while those aged 4–9 years accounted for 3% of $NO_x$ and 17% of PM concentrations. Other factors, such as environmental conditions, geographical location, and pollution sources, influence short- and long-term dispersion; further research should investigate these. Finally, the lowest estimate of d is obtained in the two presented cases for the $PM_{10}$, implying that in the event of an exogenous shock, increasing the number of emissions, the recovery will be faster with this pollutant concerning the others.

In comparing our findings with the latest studies on similarly polluted countries [61], investigated the number, frequency, and duration of pollution episodes. Their study analysed baseline air pollution trends in 100 cities, focusing on daily $PM_{2.5}$ concentrations. The results classified Delhi and Beijing into Group 1, characterised by a positive $R_{PE,norm}$ norm trend, indicating a decline in overall air pollution—while the event rate continued to rise. This suggests that although $PM_{2.5}$ concentrations are decreasing, the frequency and severity of pollution episodes are increasing, consistent with our findings. Furthermore, using data on spatiotemporal variations and trends, [62] and [63] observed a decreasing trend in $SO_2$

concentrations. However, $NO_2$ trends showed minimal improvement (−0.45±2.0 μg/m³/year), significantly lower than the reduction observed for $SO_2$ and $PM_{2.5}$ in China. These findings align with the results of our study, further supporting the air quality trends observed.

Kazakhstan has geographical conditions similar to Mongolia's and exhibits significant air pollution levels [64]. reported that $PM_{10}$ and $NO_2$ concentrations in Kazakhstan exceed WHO air quality guidelines by 2 and 6.8 times, respectively, in other countries with comparable geography and pollution levels. Their study emphasised the need for continued research and monitoring to understand air quality trends better. However, we identified a gap in defining future air pollution trends in Kazakhstan. The methodology used in this study could serve as a valuable resource for future research in this region to establish clearer projections.

## 6. Conclusions

In this paper, we have examined four pollutants ($NO_2$, $SO_2$, $PM_{10}$, and $PM_{2.5}$) in Ulaanbaatar, Mongolia, with daily data from January 1st, 2022 until May 20th, 2024 using fractionally integrated methods. The model incorporates a linear time trend to investigate its long-term pattern.

The results indicate that the time trend is statistically significantly negative in the cases of $SO_2$ and $PM_{2.5}$ implying a systematic reduction in the number of emissions across time. However, this pattern is not observed in the case of $NO_2$ and $PM_{10}$. On the other hand, referring to the differencing parameter, we see that the value of this parameter in the four series is within the interval (0, 1) and close to 0.5, which is precisely the boundary between stationary and nonstationary cases. Moreover, the fact that it is smaller than 1 implies support for mean reversion and transitory effects of exogenous shocks in the series.

This paper can be extended in several directions. Firstly, robustness checking on the results presented can be elaborated by using other parametric or even semiparametric methods. Some initial investigation, based on [65] maximum likelihood method in the time domain or the semiparametric log-periodogram approach of [66–68]), produced results that though quantitatively might differ in some cases, qualitatively were very similar, supporting the hypothesis of fractional integration in all cases. Also, from a methodological viewpoint, non-linear structures can be considered. The linear trends used in this work can be replaced by non-linear polynomials in time such as those based on Chebyshev polynomials in time and used in [69] or by Fourier functions [70] or even neural networks [71]. It would be interesting to determine if the same conclusions as those reported in this work hold under these different assumptions. From an empirical viewpoint, the analysis can be extended to other big cities all over the world. Work in these directions is now in progress.

## Supporting information

**S1 Dataset.**
(XLSX)

## Author contributions

**Conceptualization:** Chultem Batbold, Sonomdagva Chonokhuu.

**Data curation:** Chultem Batbold, Luis Alberiko Gil-Alana.

**Formal analysis:** Ariundelger Ariunsaikhan, Sonomdagva Chonokhuu, Luis Alberiko Gil-Alana.

**Investigation:** Sonomdagva Chonokhuu.

**Methodology:** Ariundelger Ariunsaikhan.

**Resources:** Ariundelger Ariunsaikhan, Chultem Batbold, Sonomdagva Chonokhuu.

**Software:** Luis Alberiko Gil-Alana.

**Supervision:** Chultem Batbold, Luis Alberiko Gil-Alana.

**Visualization:** Chultem Batbold.

**Writing – original draft:** Ariundelger Ariunsaikhan, Luis Alberiko Gil-Alana.

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
