## [Decision Letter · Decision Letter 0]

12 Feb 2025

Dear Dr. Gil-Alana,

Thank you for submitting your manuscript to PLOS ONE. After careful consideration, we feel that it has merit but does not fully meet PLOS ONE’s publication criteria as it currently stands. Therefore, we invite you to submit a revised version of the manuscript that addresses the points raised during the review process.

We look forward to receiving your revised manuscript.

Kind regards,

Naranjargal Dashdorj

Academic Editor

PLOS ONE

Journal Requirements:

https://ddfv.ufv.es/rest/api/core/bitstreams/827ad7ae-0e26-4763-8a14-303d7b4d8968/content

In your revision ensure you cite all your sources (including your own works), and quote or rephrase any duplicated text outside the methods section. Further consideration is dependent on these concerns being addressed.

5. Thank you for stating the following in your Competing Interests section: N/A. 

Reviewers' comments:

Reviewer's Responses to Questions

**Comments to the Author**

1. Is the manuscript technically sound, and do the data support the conclusions?

Reviewer #1: Yes

Reviewer #2: Partly

2. Has the statistical analysis been performed appropriately and rigorously?

Reviewer #1: Yes

Reviewer #2: I Don't Know

3. Have the authors made all data underlying the findings in their manuscript fully available?

Reviewer #1: No

Reviewer #2: Yes

4. Is the manuscript presented in an intelligible fashion and written in standard English?

Reviewer #1: Yes

Reviewer #2: Yes

Reviewer #1: 1. Why only 4 pollutants are taken this should be addressed properly as by EPA norms 6 pollutants should be present in any kind of analysis.

2. Limited literature review- Literature Review has to be given in a more detailed way. To make it convenient for others to read papers with concise structures, please summarize some related papers, for example:

https://ieeexplore.ieee.org/abstract/document/10363642

https://link.springer.com/article/10.1007/s11869-015-0369-9

https://ieeexplore.ieee.org/abstract/document/10529960

https://sensors.myu-group.co.jp/sm_pdf/SM3362.pdf

https://link.springer.com/article/10.1007/s11270-024-07012-9

https://ieeexplore.ieee.org/document/10706023

https://ieeexplore.ieee.org/document/10806566

3. Comparison with the state of the art literature is not present, newer publications should be compared

Minor comments are as follows:

1. Inconsistent Terminology: The manuscript uses different terms for similar concepts which could confuse readers.

2. Insufficient Justification for air pollutants.

3. Lack of Future Work Discussion: The manuscript does not outline potential future research directions or applications of the findings, which could enhance its impact.

4. Figures and Tables Not Referenced: Several figures and tables are mentioned but not adequately referenced in the text, making it hard for readers to follow the results.

7. Inconsistent Data Presentation: The presentation of data in figures and tables lacks uniformity, making it harder to interpret the results effectively.

Reviewer #2: This manuscript describes the behavior of air pollutants over an extended period of time and concludes that at least two of them have significantly reduce their levels. It is of mostly local relevance, but the fact that it focuses on an area that it is not well described yet, it becomes of general interest.

However, I feel the manuscript right now is not sharing important information regarding the origin and quality of the data set presented. It is analyzing secondary data without showing proper validation. An extended analysis of the location of sampling sites is also needed. Additionally, I recommend extending the conclusions to infer about nearby locations (or similar locations worldwide). Thus, the manuscript will have more global relevance instead of just being useful for local readers.

some comments are included:

Please include a description of how the data was generated in the first place, it is important to locate the reader on the type of study presented.

Section 2 of the manuscript is interesting, but it needs to clarify what kind of methods were used in each study. Only then, the comparison makes more sense. I would put special focus on the sample collection/recording methods.

Also, please describe briefly what fractional integration is.

What is the origin of the data shown in table 1? Please describe. Where was it recorded? By using what kind of instruments? Was it validated?

Define what white noise errors are

Line 69: is there any evidence of that? Or is it just the authors opinion?

Line 75: are these values measured by a state-run network? Or were they published elsewhere?

**Do you want your identity to be public for this peer review?** For information about this choice, including consent withdrawal, please see our Privacy Policy

Reviewer #1: No

Reviewer #2: No

---

## [Author Response · Author response to Decision Letter 0]

13 Mar 2025

Reviewer #1:

1.) Why only 4 pollutants are taken? This should be addressed properly as by EPA norms, 6 pollutants should be present in any kind of analysis.

Reply: Thanks for the comment.

Our study focuses on primary pollutants in Mongolia, where coal combustion is the primary source of air pollution due to its widespread use in heating systems. Other studies similar to ours either focus on analysing the spatiotemporal characteristics of primary air pollutants or on assessing air pollutants focused on these four pollutants. For instance, Zeng et al. (2019) evaluated the effectiveness of energy policies using spatial econometric methods, focusing on PM10, PM2.5, and SO2. Tian et al. (2023) examined major air pollutants' spatiotemporal and sectoral distribution and their drivers, selecting NOx, SO2, and dust as key pollutants. Similarly, Priya & Sathya (2019) focused on SO2, NO2, and PM in India, while Lau et al. (2024) analysed PM2.5 and NO2 using citywide air monitoring data from New York, USA. Other studies have adopted geospatial analysis to assess pollution trends. Sidharthan et al. (2025) monitored SO2, NO2, and PM using GIS modelling in India. Singh et al. (2023) analysed monthly data from five different monitoring stations in India, covering NO2, SO2, and PM₁₀ from 2011 to 2020. Li et al. (2020) investigated the spatiotemporal characteristics of PM2.5, PM10, SO2, and NO2 across 11 stations in China. Other relevant studies include Mulgeta et al. (2024), who analysed the spatial distribution of PM2.5 and PM10 in Addis Ababa, Ethiopia. Li et al. (2015) focused on identifying the main contributors to air pollution in Beijing, specifically analysing SO₂ levels. Sharma et al. (2019) examined air pollution trends across various geographical locations in India, considering SO2, NOx, and PM2.5. Additionally, Razy-Yanuv et al. (2022) updated previous marine air pollution estimations by analysing emissions of NOx, SOx, and PM2.5.

The above comments have now been added at the beginning of the Literature Review section.

Also, air quality monitoring follows WHO guidelines; however, not all pollutants are consistently measured in Mongolia. Some monitoring stations record only three pollutants—PM10, SO2, and NO2 —while others measure four, depending on equipment availability. Only a few stations monitor all six major pollutants. The selected monitoring site for this study, located in central Ulaanbaatar, measures four pollutants: PM10, PM2.5, SO2, and NO2. Consequently, these four variables were used in the analysis.

Dealing with this point, we have added the following comment:

“ … The air pollution pollutants in Ulaanbaatar, Mongolia, were officially measured by the National Agency for Meteorology and Environmental Monitoring (NAMEM) at 19 monitoring locations. The daily data used in this study was obtained from NAMEM or a state-run network. According to the World Health Organization (WHO), the major air pollutants include PM2.5, PM10, O₃, NO₂, SO₂, and CO. In Mongolia, air quality monitoring follows WHO guidelines; however, not all pollutants are consistently measured. Some monitoring stations record only three pollutants—PM10, SO₂, and NO₂—while others measure four, depending on equipment availability. Only a few monitoring stations measure all six major pollutants. The selected monitoring station, UB-2, is located in the center of Ulaanbaatar city. This site measures four pollutants: PM10, PM2.5, SO₂, and NO₂. Therefore, due to this limitation and justification, these four variables were used in the analysis. …”.

2.) Limited literature review- Literature Review has to be given in a more detailed way. To make it convenient for others to read papers with concise structures, please summarize some related papers, for example:

https://ieeexplore.ieee.org/abstract/document/10363642

https://link.springer.com/article/10.1007/s11869-015-0369-9

https://ieeexplore.ieee.org/abstract/document/10529960

https://sensors.myu-group.co.jp/sm_pdf/SM3362.pdf

https://link.springer.com/article/10.1007/s11270-024-07012-9

https://ieeexplore.ieee.org/document/10706023

https://ieeexplore.ieee.org/document/10806566

Reply: Many thanks for letting us know about all these interesting papers. Accordingly, we have read them and introduced some of them in the literature review, which seems now much more complete. In addition, we have also included the papers mentioned in point 1) above. Dealing specifically with the above papers, we have added the following:

“ … A Simple Linear Regression (SLR) model was used by Rabuan et al. (2023) to evaluate the accuracy of five low-cost air quality sensors against a particulate reference analyzer. Their findings suggest that low-cost sensors are unreliable for accurately measuring air quality in indoor environments. Various studies have also applied machine learning (ML) and artificial neural network (ANN) models. Borah et al. (2023) introduced a location-invariant air pollution prediction model with strong geographic generalizability, integrating light gradient-boosted regression (GBR) within a ML framework. Experiments on diverse datasets demonstrated superior performance to standard forecasting methods, such as recurrent neural networks and transformers. An interpretability analysis identified key factors influencing air pollution levels and revealed geographical patterns of high pollutant concentrations. Furthermore, Borah et al. (2024) predicted the concentrations of six major air pollutants using ML and deep learning techniques, while Patra et al. (2016) applied an ANN model to predict dust concentration in India’s deepest opencast copper mines, achieving strong agreement between observed and predicted data. …”.

3.) Comparison with the state of the art literature is not present, newer publications should be compared

Reply: We have expanded our discussion section by comparing findings with newer studies on air pollution, such as focusing on comparing air pollution trends with recent studies in similar countries from the perspective of locations and how our study contributes new insights compared to previous works. We have added the following comments in the discussion section:

“ … In comparing our findings with the latest studies on similarly polluted countries, Morawska et al. (2021) investigated the number, frequency, and duration of pollution episodes. Their study analysed baseline air pollution trends in 100 cities, focusing on daily PM2.5 concentrations. The results classified Delhi and Beijing into Group 1, characterised by a positive RPE,norm norm trend, indicating a decline in overall air pollution—while the event rate continued to rise. This suggests that although PM2.5 concentrations are decreasing, the frequency and severity of pollution episodes are increasing, consistent with our findings. Furthermore, using data on spatiotemporal variations and trends, Wang et al. (2021) and Maji & Sarkar (2020) observed a decreasing trend in SO2 concentrations. However, NO2 trends showed minimal improvement (−0.45 ± 2.0 μg/m³/year), significantly lower than the reduction observed for SO2 and PM2.5 in China. These findings align with the results of our study, further supporting the air quality trends observed.

Kazakhstan has geographical conditions similar to Mongolia's and exhibits significant air pollution levels. Kerimray et al. (2020) reported that PM10 and NO2 concentrations in Kazakhstan exceed WHO air quality guidelines by 2 and 6.8 times, respectively, in other countries with comparable geography and pollution levels. Their study emphasised the need for continued research and monitoring to understand air quality trends better. However, we identified a gap in defining future air pollution trends in Kazakhstan. The methodology used in this study could serve as a valuable resource for future research in this region to establish clearer projections. …”.

Minor comments are as follows:

1.) Inconsistent Terminology: The manuscript uses different terms for similar concepts which could confuse readers.

Reply: Thanks.

Line 73: PM2.5 levels changed to PM2.5 concentration

Line 94: SO2 levels changed to SO2 concentration

Line 111: PM levels changed to PM concentrations

Line 303: Emission of NOx changed to the concentration of NOx

Line 305: NOx emission changed to NOx concentration; PM emission changed to PM concentration

Line 306-Line 308: All the “emission” words changed to “concentration”.

2.) Insufficient Justification for air pollutants.

Reply: Thanks for the comment. We have provided a justification for selecting four air pollutants in response to Major Comment 1. In addition, we would like to elaborate on the source-specific rationale for this selection. The primary sources of air pollution in Ulaanbaatar, Mongolia, are household heating systems, power plants (both reliant on coal combustion), and transportation. Numerous studies have examined the sources of air pollution in Ulaanbaatar and have consistently identified household emissions as the dominant contributor, followed by power plant emissions and transportation-related pollution. The following references support this conclusion:

Batmunkh, T., Kim, Y. J., Jung, J. S., Park, K., & Tumendemberel, B. (2013). Chemical characteristics of fine particulate matters measured during severe winter haze events in Ulaanbaatar, Mongolia. Journal of the Air & Waste Management Association, 63(6), 659-670. https://doi.org/10.1080/10962247.2013.776997

Davy, P. K., Gunchin, G., Markwitz, A., Trompetter, W. J., Barry, B. J., Shagjjamba, D., & Lodoysamba, S. (2011). Air particulate matter pollution in Ulaanbaatar, Mongolia: determination of composition, source contributions and source locations. Atmospheric Pollution Research, 2(2), 126-137. DOI:10.5094/APR.2011.017

Government of Mongolia, Swiss Agency for Development and Cooperation SDC, UNICEF. (2023). Impact of air pollution on maternal and child health project. p10. Available online: https://www.unicef.org/mongolia/media/5071/file/Impact%20of%20Air%20Pollution%20on%20Maternal%20and%20Child%20Health%20Project%20(2018-2023).pdf (accessed on 18 February 2025)

Guttikunda, S. K., Lodoysamba, S., Bulgansaikhan, B., & Dashdondog, B. (2013). Particulate pollution in Ulaanbaatar, Mongolia. Air Quality, Atmosphere & Health, 6, 589-601. DOI: 10.1007/s11869-013-0198-7

The Air Pollution Reducing Department of Capital City (APRD). (2023). Source Emissions Inventory Annual Report 2021. p46-50 Available online: https://aprd.ub.gov.mn/index.php?newsid=729 (accessed on 18 February 2025)

The Air Pollution Reducing Department of Capital City (APRD)., Japan International Cooperation Agency (JICA). (2017), Capacity Development Project for Air Pollution Control in Ulaanbaatar City Phase 2 in Mongolia Final Report. p147-159 Available online: https://openjicareport.jica.go.jp/pdf/12289310.pdf (accessed on 18 February 2025)

Zinicovscaia, I., Narmandakh, J., Yushin, N., Peshkova, A., Chaligava, O., Tsendsuren, T. O., ... & Tsogbadrakh, T. (2024). Assessment of Air Pollution in Ulaanbaatar Using the Moss Bag Technique. Archives of Environmental Contamination and Toxicology, 86(2), 152-164. https://doi.org/10.1007/s00244-024-01050-4

See the new paragraph on the Introduction (page 2) explaining this point further.

3.) Lack of Future Work Discussion: The manuscript does not outline potential future research directions or applications of the findings, which could enhance its impact.

Reply: Thanks for the comment. We have extended the final paragraph in the conclusions that reads now as follows:

“ … Firstly, robustness checking of the results presented can be elaborated by using other parametric or even semiparametric methods. Some initial investigation, based on Sowell’s (1992) maximum likelihood method in the time domain or the semiparametric log-periodogram approach of Geweke and Porter-Hudak (GPH, 1983; Robinson, 1995; Shimotsu and Phillips, 2002), produced results that though quantitatively might differ in some cases, qualitatively were very similar, supporting the hypothesis of fractional integration in all cases. Also, from a methodological viewpoint, non-linear structures can be considered. The linear trends used in this work can be replaced by non-linear polynomials in time such as those based on Chebyshev polynomials in time and used in Cuestas and Gil-Alana (2016) or by Fourier functions (Gil-Alana and Yaya, 2021) or even neural networks (Yaya et al., 2021). It would be interesting to determine if the same conclusions as those reported in this work hold under these different assumptions. From an empirical viewpoint, the analysis can be extended to other big cities all over the world. Work in these directions is now in progress. …”.

4.) Figures and Tables Not Referenced: Several figures and tables are mentioned but not adequately referenced in the text, making it hard for readers to follow the results.

Reply: Thanks. All figures and tables are now referenced.

7.) Inconsistent Data Presentation: The presentation of data in figures and tables lacks uniformity, making it harder to interpret the results effectively.

Reply: Thanks. We have better explained the meaning of tables and figures in the revised version of the paper.

Other minor changes

Title page: Comments from the Editor and two anonymous reviewers are gratefully acknowledged.

*.) We have corrected several typos and grammatical errors all over the manuscript.

Line 158: We have replaced Naveen (2017) by Naveen and Anu (2017).

Tables 3 and 5: We have explained in an endnote that --- means lack of statistical significance.

References

Batmunkh, T., Kim, Y. J., Jung, J. S., Park, K., & Tumendemberel, B. (2013). Chemical characteristics of fine particulate matters measured during severe winter haze events in Ulaanbaatar, Mongolia. Journal of the Air & Waste Management Association, 63(6), 659-670. https://doi.org/10.1080/10962247.2013.776997

Cuestas J.C. & Gil-Alana, L.A. (2016). A Non-Linear Approach with Long Range Dependence Based on Chebyshev Polynomials. Studies in Nonlinear Dynamics and Econometrics, 23, 445–468.

Davy, P. K., Gunchin, G., Markwitz, A., Trompetter, W. J., Barry, B. J., Shagjjamba, D., & Lodoysamba, S. (2011). Air particulate matter pollution in Ulaanbaatar, Mongolia: determination of composition, source contributions and source locations. Atmospheric Pollution Research, 2(2), 126-137. DOI:10.5094/APR.2011.017

Ethiopia. Stochastic Environmental Research and Risk Assessment, 1-19.

Geweke, J. and S. Porter-Hudak (1983), The estimation and applicatons of long memory time series models, Journal of Time Series Analysis 4(4), 221-238.

Gil-Alana, L.A. & Yaya, O. (2021). Testing fractional unit roots with non-linear smooth break approximations using Fourier functions. Journal of Applied Statistics, 48, 13-15, 2542-2559.

Guttikunda, S. K., Lodoysamba, S., Bulgansaikhan, B., & Dashdondog, B. (2013). Particulate pollution in Ulaanbaatar, Mongolia. Air Quality, Atmosphere & Health, 6, 589-601. DOI: 10.1007/s11869-013-0198-7

Kerimray, A., Assanov, D., Kenessov, B., & Karaca, F. (2020). Trends and health impacts of major urban air pollutants in Kazakhstan. Journal of the Air & Waste Management Association, 70(11), 1148-1164. https://doi.org/10.1080/10962247.2020.1813837

Lau, K., Guo, J., Miao, Y., Ross, Z., Riley, K. W., Wang, S., ... & Perera, F. (2024). Major air pollution and climate policies in NYC and trends in NYC air quality 1998–2021. Frontiers in Public Health, 12, 1474534.

Li, S., Feng, K., & Li, M. (2017). Identifying the main contributors of air pollution in Beijing. Journal of Cleaner Production, 163, S359-S365.

Li, L., Zhao, Z., Wang, H., Wang, Y., Liu, N., Li, X., & Ma, Y. (2020). Concentrations of four major air pollutants among ecological functional zones in Shenyang, Northeast China. Atmosphere, 11(10), 1070.

Maji, K. J., & Sarkar, C. (2020). Spatio-temporal variations and trends of major air pollutants in China during 2015–2018. Environmental Science and Pollution Research, 27, 33792-33808. https://doi.org/10.1007/s11356-020-09646-8

Morawska, L., Zhu, T., Liu, N., Torkmahalleh, M. A., de Fatima Andrade, M., Barratt, B., ... & Ye, C. (2021). The

---

## [Decision Letter · Decision Letter 1]

2 Apr 2025

ATMOSPHERIC POLLUTION IN ULAANBAATAR. PERSISTENCE AND LONG-RUN TRENDS

PONE-D-24-53171R1

Dear Dr. Gil-Alana,

We’re pleased to inform you that your manuscript has been judged scientifically suitable for publication and will be formally accepted for publication once it meets all outstanding technical requirements.

Kind regards,

Naranjargal Dashdorj

Academic Editor

PLOS ONE

Comments from PLOS Editorial Office:

We note that one or more reviewers has recommended that you cite specific previously published works in an earlier round of revision. As always, we recommend that you please review and evaluate the requested works to determine whether they are relevant and should be cited. It is not a requirement to cite these works and you may remove them before the manuscript proceeds to publication. We appreciate your attention to this request.

Reviewers' comments:

Reviewer's Responses to Questions

**Comments to the Author**

Reviewer #1: All comments have been addressed

Reviewer #2: All comments have been addressed

2. Is the manuscript technically sound, and do the data support the conclusions?

Reviewer #1: Yes

Reviewer #2: Yes

3. Has the statistical analysis been performed appropriately and rigorously?

Reviewer #1: Yes

Reviewer #2: Yes

4. Have the authors made all data underlying the findings in their manuscript fully available?

Reviewer #1: Yes

Reviewer #2: Yes

5. Is the manuscript presented in an intelligible fashion and written in standard English?

Reviewer #1: Yes

Reviewer #2: Yes

Reviewer #1: Authors are suggested to compare their result with the 2025 publications like as follows:

1. https://ieeexplore.ieee.org/abstract/document/10706023

2. https://ieeexplore.ieee.org/document/10806566

The references should be related to the air pollutants prediction only

Reviewer #2: (No Response)

**Do you want your identity to be public for this peer review?** For information about this choice, including consent withdrawal, please see our Privacy Policy

Reviewer #1: No

Reviewer #2: No

---

## [Editor Report · Acceptance letter]

PONE-D-24-53171R1

PLOS ONE

Dear Dr. Gil-Alana,

I'm pleased to inform you that your manuscript has been deemed suitable for publication in PLOS ONE. Congratulations! Your manuscript is now being handed over to our production team.

Kind regards,

on behalf of

Dr. Naranjargal Dashdorj

Academic Editor

PLOS ONE